# Genome-Wide Characterization of Effector Protein-Encoding Genes in *Sclerospora graminicola* and Its Validation in Response to Pearl Millet Downy Mildew Disease Stress

**DOI:** 10.3390/jof9040431

**Published:** 2023-03-31

**Authors:** Shiva Hadimani, Savitha De Britto, Arakere C. Udayashankar, Nagaraj Geetha, Chandra S. Nayaka, Daoud Ali, Saud Alarifi, Shin-ichi Ito, Sudisha Jogaiah

**Affiliations:** 1Laboratory of Plant Healthcare and Diagnostics, PG Department of Biotechnology and Microbiology, Karnatak University, Dharwad 580003, India; 2Division of Biological Sciences, School of Science and Technology, University of Goroka, Goroka 441, Papua New Guinea; 3Department of Studies in Biotechnology, University of Mysore, Manasagangotri, Mysuru 570006, India; 4Department of Zoology, College of Science, King Saud University, P.O. Box 2455, Riyadh 11451, Saudi Arabia; 5Research Center for Thermotolerant Microbial Resources, Yamaguchi University, Yamaguchi 753-8515, Japan; 6Department of Environmental Science, Central University of Kerala, Tejaswini Hills, Periye (PO) 671316, Kasaragod (DT), Kerala, India

**Keywords:** *Sclerospora graminicola*, pearl millet, effectors, crinkler effectors, RxLR effectors, RxLR-dEER effectors, downy mildew, signal peptide, secretory proteins

## Abstract

Pearl millet [*Pennisetum glaucum* (L.) R. Br.] is the essential food crop for over ninety million people living in drier parts of India and South Africa. Pearl millet crop production is harshly hindered by numerous biotic stresses. *Sclerospora graminicola* causes downy mildew disease in pearl millet. Effectors are the proteins secreted by several fungi and bacteria that manipulate the host cell structure and function. This current study aims to identify genes encoding effector proteins from the *S. graminicola* genome and validate them through molecular techniques. In silico analyses were employed for candidate effector prediction. A total of 845 secretory transmembrane proteins were predicted, out of which 35 proteins carrying LxLFLAK (Leucine–any amino acid–Phenylalanine–Leucine–Alanine–Lysine) motif were crinkler, 52 RxLR (Arginine, any amino acid, Leucine, Arginine), and 17 RxLR-dEER putative effector proteins. Gene validation analysis of 17 RxLR-dEER effector protein-producing genes was carried out, of which 5genes were amplified on the gel. These novel gene sequences were submitted to NCBI. This study is the first report on the identification and characterization of effector genes in *Sclerospora graminicola*. This dataset will aid in the integration of effector classes that act independently, paving the way to investigate how pearl millet responds to effector protein interactions. These results will assist in identifying functional effector proteins involving the omic approach using newer bioinformatics tools to protect pearl millet plants against downy mildew stress. Considered together, the identified effector protein-encoding functional genes can be utilized in screening oomycetes downy mildew diseases in other crops across the globe.

## 1. Introduction

Effectors are proteins secreted by several fungi and bacteria that manipulate the host cell structure and function. They are reported to cause infection or induce defense responses in the host [1,2]. This contradictory nature of effectors has been encountered in many fungal and bacterial plant diseases [3,4]. Depending on where they are found inside the host plant, effectors are categorized into two types: cytoplasmic and apoplastic. In the plant extracellular spaces, apoplastic effectors are released, whereas cytoplasmic effectors, on the other hand, are discharged within the plant cytoplasm via the pathogen’s specialized structures, such as haustoria and vesicles. The delivery methods of effectors in fungi and oomycetes are yet unknown [5]. Effector delivery has been linked to conserved sequence motifs in the sequence divergent oomycete effectors [6]. The classification of fungal effectors based on sequence characteristics and conserved motifs poses a major challenge [7]. A ‘pathogen’s effectome (sometimes referred to as effectorome) is the repertoire of all its effector proteins [8]. The cytoplasmic RxLR effectors consist of conserved N-terminal 4 amino acids, RxLR (Arginine, any amino acid, Leucine, Arginine) motif [9], followed by a dEER (Aspartic acid, Glutamic acid, Glutamic acid-Arginine) motif [10,11,12]. The crinkler cytoplasmic effector contains LxLFLAK (Leucine–any amino acid–Phenylalanine–Leucine–Alanine–Lysine) motif [12,13,14]. The migration of effectorsfurther into a plant is found to be dependent on these motifs [15]. In the Peronosporales clade, which includes downy mildew pathogens, numerous RxLR effectors were reported [16,17]. The intrinsic disorder of oomycete RxLR protein is a typical feature [18,19]. Several evolutionary mechanisms that can drive RxLR effector diversity within Peronosporales have been documented, including gene recombination, duplication events, and point mutations [20,21]. Although sequence motif and Hidden Markov Model searches are well-established approaches in oomycetes for predicting certain classes of cytoplasmic effectors, they overlook genuine effectors without such motifs, domains, or apoplastic effectors [22].

Plant diseases caused by fungus and oomycetes are devastating. These eukaryotic filamentous pathogens produce effector proteins that infect the plant body. The fungal and oomycete pathogens have different infection methods, and their effectors differ considerably in sequence homology. However, they share similar host habitats, plant apoplasts, or cytoplasms and hence may share some unifying qualities based on the host compartments’ requirements [23]. A typical example of an oomycete pathogen is *Sclerospora graminicola*, which causes downy mildew (green ear) disease in pearl millet [24] with 20–80% yield loss [25]. Downy mildew occurrence has been moderately adaptable on diverse hybrids, and more than 90% incidence has been noted on some crosses in ‘farmers’ fields [26,27]. The spread of downy mildew disease is favored by high relative humidity (85–90%) with moderate temperatures (20–30 °C). They are included in a heterogeneous group of obligatory biotrophs that infect economically important crops such as grapevines [28], sorghum [29], and pearl millet [25]. These physiognomies of the fungus make it exceptionally flexible and adjustable to varied environmental situations [30,31].

Despite the economic importance of the disease caused by *Sclerospora graminicola*, the functional annotation of the genome has not been documented, and their effector report is unknown. Additionally, there is a lack of literature on the role of effector proteins in *S. graminicola* and its expression data. Hence, the current investigation was carried out to identify and characterize the candidate effector proteins of *S. graminicola* by data mining and computational analysis and validation through molecular techniques.

## 2. Materials and Methods

### 2.1. GeneMark-ES Suite

The draft genome of *Sclerospora graminicola* was recovered from https://www.ncbi.nlm.nih.gov/bioproject/PRJNA325098/ (accessed on 27 August 2021) to predict genes and proteins using the GeneMark-ES suite. The ES and fungus flags were used with GeneMark script to enable self-training and branch point model to predict genes with default parameters, and the following measures were used to include the sequence containing the gene in the test set: a. The gene must have an initiator codon ATG, a conical acceptor/donor site; b. Intron/exon assembly must be reinforced by expressed sequence tag/complementary deoxyribonucleic acid [32,33]; c. The annotationdoes have to include substitute isoforms accompanied by EST/cDNA; d. There must be no gene overlap with any other genes that have been annotated; e. Multiple-gene sequences are more suitable for precision evaluation [34,35].

### 2.2. Identification of Signal Peptides (SP) in the N-Terminal Region

SignalP 6.0 (services.healthtech.dtu.dk/services/SignalP-6.0/) (accessed on 23 March 2022) was used to detect signal peptides (SP) in the N-terminal region. The amino acid sequence was converted into FASTA format and pasted in the given empty box given. Furthermore, appropriate options were selected, and the command line “signalpinput.fasta” was submitted. The results showed predicted SP and the position of the cleavage site [36,37,38].

### 2.3. Target P Server

The sequences obtained from SignalP 6.0 were evaluated by TargetP v1.1 (http://www.cbs.dtu.dk/services/TargetP/) (accessed on 1 December 2021) [39] for their sub-cellular location based on N-terminal pre-sequences (at least the first 130 amino acids of the N-terminus required). The input data was a one-letter amino acid code reset; other symbols got converted to X before processing, and a non-plant option was selected before submitting the input [40,41,42].

### 2.4. TMHMM v2.0

The input sequences were in FASTA format with functional and secretory pathway proteins, and signal peptides were checked for the presence of transmembrane domains by TMHMM v2.0 (https://services.healthtech.dtu.dk/service.php?TMHMM-2.0) (accessed on 3 January 2022) [43,44,45]. Proteins with 0 and 1 TM domains (an N-terminal signal peptide) were combined to get the secretome of *Sclerospora graminicola*. Further, LxLFLAK and RxLR motifs were searched in secretome proteins using pattern matches to identify crinkler (CRN) and RxLR proteins, and all the results were cross verified with EffectorP 3.0 (http://effectorp.csiro.au/) tool (accessed on 27 April 2022) [23]. The protein sequences were translated to their respective gene sequence, and a similarity search was carried out using the Basic Local Alignment Search Tool (BLAST) of NCBI (https://blast.ncbi.nlm.nih.gov/) (accessed on 18 November 2022).

### 2.5. Host and Pathogen

The downy mildew pathogen, *Sclerospora graminicola*, was isolated from a highly susceptible pearl millet host cultivar (7042S) grown in earthen pots (12–15 cm diameter) under greenhouse conditions. The pathogen was maintained on the same host throughout the experiment.

### 2.6. Extraction of RNA from Scelrospora Graminicola and cDNA Synthesis

The genomic DNA of the host plant was isolated from susceptible pearl millet leaves, as described by Divya et al. [46]. Leaf samples were collected from *Sclerospora graminicola*-infected pearl millet plants, rinsed with water to eliminate unwanted dirt and dust, and cleaned with sterile tissue paper. Clean leaves were stapled onto wet blotter disc and placed on the upper lid of a sterile Petri dish. The sterile Petri dish was filled with 15 mL of sterile water and incubated overnight at a temperature of 18–20 °C. During the early hours, spores were collected in the lower lid of the Petri dish and centrifuged at 5000 rpm for 5 min. Total RNA extraction processes were initiated by repeatedly washing the zoospores three times in sterile distilled water. The zoospores were washed thrice in sterile distilled water, and total RNA extraction was executed with the aid of RNAeasy plant micro kit as per manufacturer instructions (Qiagen, Hilden, Germany). Total RNA isolated from *S. graminicola* was checked for its purity at the absorbance of A260/A280 in Ultraviolet-visible spectroscopy of Agilent (Cary 60 UV-Vis). The cDNA synthesis was performed with RNA templates using oligo (dT) _18_ primers (ThermoFisher, Madison, WI, USA).

### 2.7. Primer Designing

For gene validation, PCR primers for a subset of anticipated full-length RxLR-dEER coding genes were designed and synthesized (Sigma-Aldrich Chemicals Pvt. Ltd., Bangalore, India). The designed primers used in this present study are mentioned in Table 1. Briefly, the full length of the effector protein-encoding nucleotide sequence from the genome was designed manually by selecting a few nucleotides from the site of initiation and the site of termination. A few nucleotides were selected based on the reverse complement tool. To calculate the melting temperature of the primer, we used the percent GC Oligocalc tool.

### 2.8. PCR Amplification

Polymerase Chain Reaction (PCR) experiments were carried out in a thermal cycler (C1000 Touch, part no, #1851148, BioRad, Philadelphia, PA, USA) on cDNA, and only the successfully amplified and reproducible segments were analyzed after the procedure was repeated thrice for each isolate individually. Deoxyribonucleic acid amplification was conducted in a 20 µL reaction mixture containing 0.2 mM of primer and dNTPs, 0.6 units Taq pol (Banglore Genei, Bengaluru, India), 10 mM of tris hydrochloride (pH 9.0), 1.5 mM magnesium chloride, 50 mM potassium chloride, and 50 ng DNA. PCR cycling settings were as follows: initial denaturation at 94 °C for 4 min followed by 40 cycles of 1 min at 94 °C, 1 min at primer-specific annealing temperature (Table 1), and 2min at 72 °C, with final extension for 10 min at 72 °C. The amplicons were electrophoresed on an agarose gel after adding bromophenol blue on 1.5% agarose gel stained with EtBr using 1 Tris-borate Ethylene diamine tetra-acetic acid buffer pH 8.3 [47]. A 1 kb DNA Ladder (part no: G571A, Promega Corporation, Madison, WI, USA) was used as molecular weight marker (m) at 60–65 V. The gel slab was removed and visualized under a molecular imager (Gel Doc imaging systems XR+, BIO RAD).

The amplicons were extracted from the gel with a sharp, sterile scalpel blade when the gel was illuminated with a UV-transilluminator (70%). The dissected gel fragments were added to a clean 2 mL microcentrifuge tube that had been pre-weighed. According to the technique provided, with the help of PureLink Quick Gel Extraction Kit (Cat.No.K210012, Invitrogen, Waltham, MA, USA), the required amplicon was recovered off the agarose gel, and the eluted product was subjected to Sanger sequencing (3730 DNA Analyzer 48-Capillary Array). The results were BLAST analyzed in NCBI for homology with any RxLR effector protein-encoding genes. The amplified nucleotide sequences of RxLR-dEER effectors were retrieved from the direct sequencing and converted to their respective amino acid sequences using the Translate tool, and the sequences with 5′ to 3′, which had no gap, were selected. Screening of RxLR and dEER motif was carried out manually, and the intrinsic disorder of the respective proteins was investigated based on the predicted output of the PONDAR VL-XT tool [48,49].

## 3. Results

### 3.1. Secretome Mapping

Protein sequences were obtained from Genemark-ES software and used as inputs for SignalP 6.0 to identify secretory proteins. Signal peptides, identified by SignalP 6.0, were present in 935 protein sequences. Out of 935 proteins, 911 were predicted to be involved in secretory pathway signal peptides as per TargetP v1.1.TMHMM v2.0 was used, which predicted 845 secretory proteins in which 803 proteins had 0 TM, and 42 proteins had 1 TM domain (an N-terminal signal peptide) as output. One of the requirements for classifying a protein as an effector is that it secretes extracellularly through the N-terminal secretion signal [50,51]. It was observed that in 35 proteins, LxLFLAK motifs were present, and 152 proteins had RxLR motifs.

### 3.2. Annotation for Crinklerand RxLR Effectors

Leucine–any amino acid–Phenylalanine–Leucine–Alanine–Lysine (LxLFLAK) motifs were present in 35 proteins and were labeled as crinkler (CRN) effector proteins. Furthermore, RxLR proteins were filtered based on the criteria that these motifs are present within 30–60 amino acids after signal cleavage site, and cleavage site is present within 30 amino acids using in-house pearl scripts [52,53]. This led to the identification of 69 RxLR motifs containing proteins that are designated as RxLR and further RxLR effectors carrying dEER motifs were screened, and 17 were identified based on the dEER motifs they were carrying in amino acid sequence after RxLR motifs (Figure 1).

### 3.3. EffectorP Machine Learning

All the predicted proteins were cytoplasmic effectors with 0 to 1 values. Interestingly, 4 CRN effectors (1492_g, 10105_g, 42778_g, and 53743_g) out of 35 (Table 2), 5 RxLR effectors (12504_g, 11898_g, 59897_g, 63043_g, and 64364_g) out of 52 (Table 3), and 1RxLR-dEER effector (1854_g) out of 17 were predicted as non-effector by EffectorP 3.0 tool. In addition, 7856_g (Table 4) was predicted to be a cytoplasmic effector and apoplastic effector.

### 3.4. Similarity Search Using NCBI BLAST Tool

The predicted Crinkler and RxLR nucleotide sequences were subjected to NCBI BLAST search. Out of 35 crinkler (CRN) genes, 9 had similarity with *Phytophthora sojae* strain P6479, 10 with *Plasmopara halstedii*, 3 with *Phytophthora infestans* T30-4, 3 with *Lagenidium giganteum f. caninum*, and 10 had no similarity with any of the genes in the database. Out of 52 RxLR genes, only one gene had a similarity with *Plasmopara halstedii*, the rest of the 51 had no similarity, and all 17 RxLR-dEER effectors had no similarity in the NCBI database. However, the translated RxLR protein sequence showed similarity with other proteins found in the NCBI database (Appendix A; Appendix A).

### 3.5. Confirmation of the Presence of RxLR-dEER Effectors Genes

Total RNA isolated from *S. graminicola* had a purity of 1.80 at the absorbance of A260/A280 in Cary 60 UV-Vis, an Ultraviolet-visible spectrophotometer by Agilent. The amplicons of all the 17 RxLR-dEER protein-coding genes were subjected to 1.5% agarose gel electrophoresis along with host DNA (*Pennisetum glaucum*), a ladder of 1 kb. Interestingly, only 5bands (6877_g, 60945_g, 8311_g, 35983_g, and 60741_g) were visible at 885, 1248, 1254, 1410, and 1533 base pairs, respectively (Figure 2). These five genes were BLAST analyzed in the NCBI database for homology with any RxLR effectors. All five amplicons had no significant similarity in the NCBI database. Hence, the amplified sequences were submitted to NCBI Gene Bank (Table 5).

### 3.6. Analysis of Overall Disorder Regions in RxLR-dEER Effector Proteins Using PONDR VL-XT

The nucleotide sequences of the five amplicons were translated to their respective amino acid sequences using the bioinformatics tool Translate, and the sequences were selected in the frame of 5′ to 3′endswith no gaps in it and at least having one open reading frame (ORF). The disordered content in predicted RxLR-dEER proteins ranged from 46.17% to 25.05% (Figure 3A–E), and the mean disorder content was 33.928% (Table 6). The sequence features and predicted domains of the five novel effector proteins are presented in Figure 4 and Table 7.

## 4. Discussion

Pathogens release effector proteins into the plant apoplast or transport them into the host cytoplasm, where they inhibit defense responses or change host metabolism [51,54]. The oomycete cytoplasmic RxLR and crinkler (CRN) effector classes are well documented based on the modular structure [55]. CRNs are a deep-rooted family of effectors discovered in various oomycete species with different evolutionary relationships [15]. In this present study, we discovered 104 effector-encoding genes in *S. graminicola*. A similar study carried out by Muller et al. [56] found 844 putative effector genes in *Blumeria graminis f.* sp. *tritici*. Huang et al. [57] used bioinformatic prediction approaches to identify 316 candidates secreted effector proteins (CSEPs) in the complete genome of *Fusarium sacchari*. A total of 95 CSEPs, spanning 40 superfamilies and 18 domains, had known conserved structures, while another 91 CSEPs comprised 7 recognized motifs. A total of 14 of the 130 CSEPs with no known domains or motifs had 1 of 4 unique motifs. The roles of 163 CSEPs were investigated using a heterogeneous expression system in *Nicotianabenthamiana*. In *N. benthamiana*, seven CSEPs reduced BAX-triggered programmed cell death, while four caused cell death. These eleven CSEPs’ expression characteristics during *F. sacchari* infection revealed that they could be involved in sugarcane-*F. sacchari* interaction. The *B. graminis f.* sp. *tritici*, the powdery mildew pathogen of the wheat, genome sequence revealed that it encoded 7588 proteins that coded genes in 180 Mb genomic size. *B. graminis f.* sp. *hordei* genome has 5854 protein-coding genes [58]. A total of 660 and 620 secretory proteins were recognized in 2 individual races, 77 and 106, of *Puccinia triticina*, respectively [59]. Our analysis of *S. graminicola* predicted that 845 out of 79,754 proteins are secretory trans-membrane proteins. Furthermore, out of these secretory proteins, 35 CRN effector and 69 RxLR effector proteins were identified. This is the first report on the identification and characterization of effector genes in *Sclerospora graminicola*, a downy mildew agent affecting *Pennisetum glaucum*.

From this present work, it is evident that the genome of *S. graminicola* has only thirty-five crinkler (CRN) effectors proteins present. This might be because oomycetes are known to secrete CRN without a traditional signal peptide via an unusual secretion pathway. Because CRNs are known to cause necrosis, which is unfavorable to biotrophy, the lower CRN level in downy mildew pathogens could indicate an adaptation to biotrophy [17]. Moreover, 152 RxLR motifs contained effector proteins, out of which 69 had RxLR motifs within 30–60 amino acid sequences present after the signal cleavage site and cleavage site within 30 amino acids [52] in *S. graminicola*. In the future, a better understanding of fungal effector function and the underlying mechanisms and the application of host-induced gene silencing technology to generate disease-resistant crops could be an effective method for preventing and controlling plant illnesses [60].

The pathogenicity factors of downy mildew pathogens with the N-terminal RxLR motif are the best understood. Purayannur et al. [17] reported 296 RxLR effectors in *P. humuli*. Similarly, Tyler et al. [61] discovered 350 RxLR effectors in *Phytophthora sojae* and *P. ramorum* genomes, respectively. At least fifty downy mildew effectors have been discovered in *Hyaloperonospora parasitica* by Morgan and Kamoun [62]. Kamoun [63] reported 200 effectors in *P. sojae*, *P. capsici*, *P. infestans*, and *P. ramorum*, respectively. Baxter et al. [14] found 130 genes in *H. arabidopsidis* expressing putative effector proteins with RxLR dEER motifs. In a different study, Cai et al. [64] reported that the virulence of the bacterial pathogen, *Aeromonas salmonicida* is highly dependent on the effector protein hcp gene for its pathogenicity. In this study, the hypersensitive reaction was indicated by the development of necrotic areas on the resistant pearl millet callus within 2 h post inoculation, thus triggering defense signaling responses in the neighboring cells.

This study witnessed RxLR-dEER protein 35893_g (OM365913) had the highest disordered residues of 46.17% in the secretome of *S. graminicola*. Similarly, in a different investigation, it was reported that *P. sojae* had an average of 63% of disordered amino acid residues as RxLR-dEER proteins have a unique amino acid makeup and are high in disorder-promoting residue, and the disordered structure of the effectors may boost their pathogenic ability. Thus, proteins could interface with plant proteins to imitate host defense signaling molecules and control plant physiological responses [19,65,66]. Parallel to the genomic investigations that paved the way for studying effector and pathogen genetics, transcriptome analyses on biotrophs are popular; it has also improved pathogen understanding by bringing transcriptome research and genome-slicing investigations, employing complementary DNA libraries. Because biotroph effector molecules are generally unique and have little resemblance to existing proteins, selecting candidates based solely on sequences becomes more difficult [67]. Hacquard et al. [68] discovered a multistep mode of action of mildew candidate secretory effector proteins (CSEPs) in powdery mildew pathogenesis on barley and immune compromised *Arabidopsis*, with a first wave of CSEP transcripts accumulating during host cell entry (12 h) and a second wave of transcripts accumulating at the stage of haustorium formation (24 h). In wheat powdery mildew, a comparable high induction of CSEPs was seen during the haustorial stage [69].

## 5. Conclusions

Obligate biotrophs are among the fastest developing pathogens; they might use many mechanisms driving effector development or encode a high quantity of nontypically discharged effector-producing genes. Transcripts jammed with the host nucleotide sequences may obscure effector detection as these organisms tightly govern effector regulations and analyze expression in diseased plants. Effector proteins have an extremely varied sequence, and almost no protein has a resemblance to identified effectors. In this present study, Gene mark ES suite, SignalP 6.0, TargetP, and TMHMM v.2.0 could define the ‘organism’s entire secretome from the projected proteomes, and Effector 3.0 positively correlated with the results of the above tools. This study clearly shows that the genome of *Sclerospora graminicola* comprises two classes of effectors that are RxLR and crinkler (CRN), of which five of the novel RxLR with dEER motif effector proteins are documented in the NCBI database. Furthermore, the presence of intrinsic disorder in these proteins is a unique structural property of RxLR proteins. This is the first report to document the presence of CRN, RxLR, and RxLR-dEER effector proteins in the *S. graminicola* genome. Further study on the interaction of RxLR proteins with host plants would provide a new area for confirmation of the pathogenic or mimicking activity of the protein to trespass the host immune response to cause the disease.

## Figures and Tables

**Figure 1 jof-09-00431-f001:**
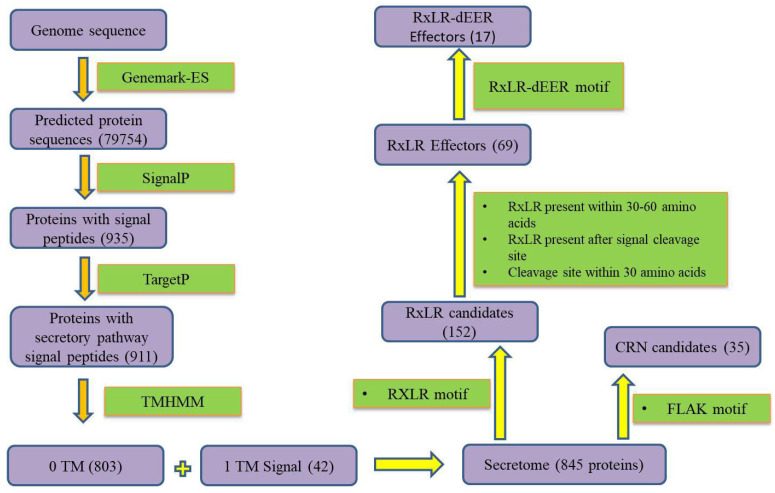
Secretome mapping and annotation for CRN and RxLR effectors.

**Figure 2 jof-09-00431-f002:**
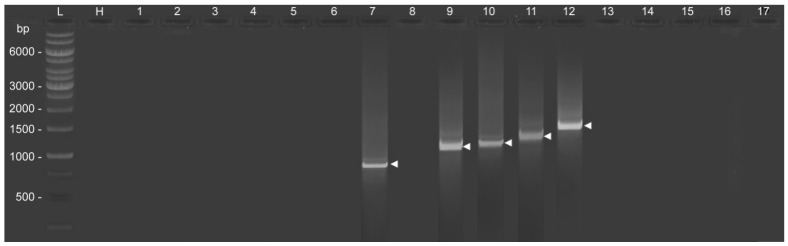
The fingerprint of 17 PCR amplicons (1 to 17) and Host (H) (*Pennisetum glaucum*) on 1.5% agarose gel electrophoresis, out of which only 5 bands with base pair size 885 (lane 7), 1248 (lane 9), 1254 (lane 10), 1410 (lane 11), and 1533(lane 12) are visible. The lane marked L represents a 1 kb ladder.

**Figure 3 jof-09-00431-f003:**
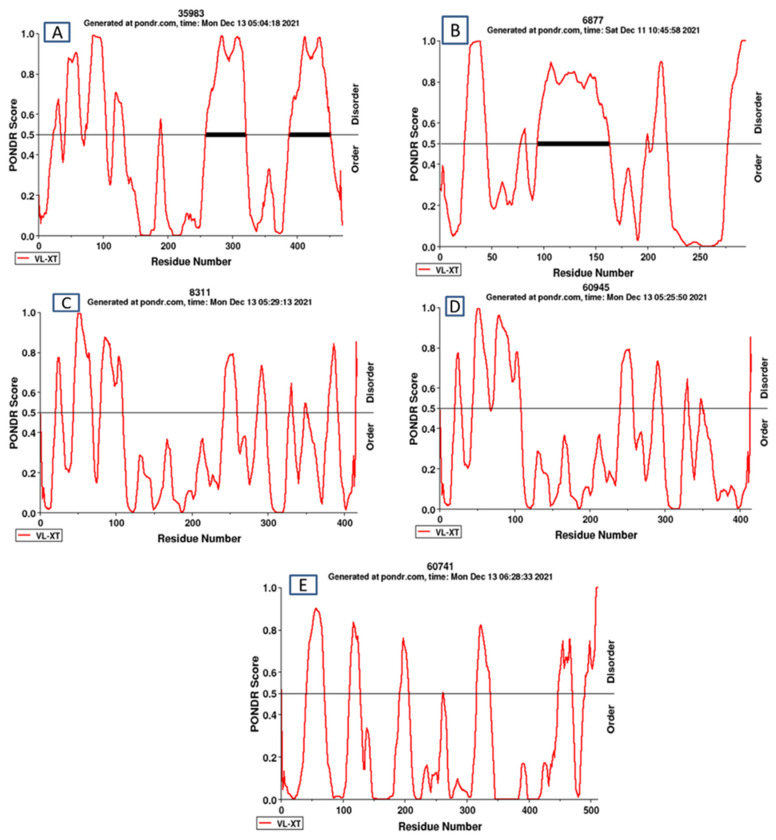
RxLR-dEER effector proteins showing the disorder regions. (**A**). Sequence 35983_g with overall 46.17% disordered amino acid residues. (**B**). Sequence 6877_g with overall 43.88% disordered amino acid residues. (**C**). Sequence8311_g with overall 28.06% disordered amino acid residues. (**D**). Sequence60945_g with overall 26.75% disordered amino acid residues. (**E**). Sequence 60741_g with overall 25.05% disordered amino acid residues.

**Figure 4 jof-09-00431-f004:**
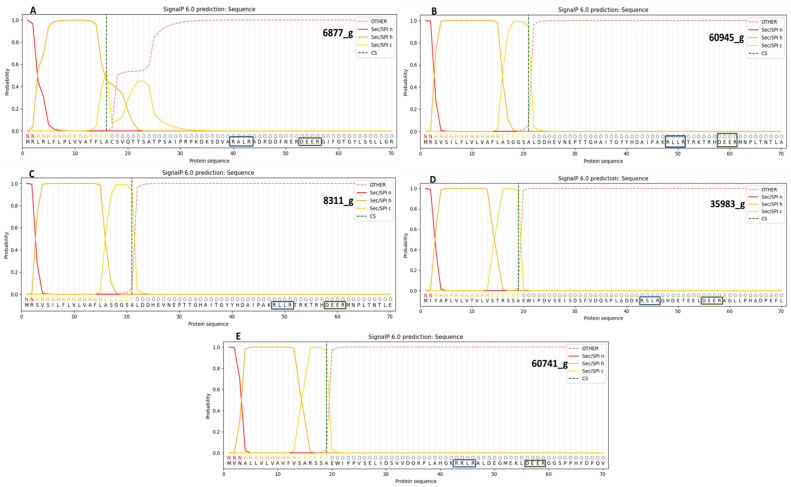
Effector proteins representing the secretary signal peptide and its cleavage site generated using SignalP 6.0. (**A**). One secretory signal peptide exists in protein 6877_g, and its cleavage site is located at position 16 with a probability of 0.9996. (**B**). A secretory signal peptide with cleavage site at 21st position, and a probability of 0.9997 is present in the protein 60945_g. (**C**). A secretory signal peptide and a cleavage site, both with a probability of 0.9997, are present in protein 8311_g. (**D**). Protein 35983_g has one secretory signal peptide and cleavage site at 19th position with likelihood of 0.9998. (**E**). A secretory signal peptide and cleavage site are present in 60741_g at position 19 with likelihood of 0.9998.

**Table 1 jof-09-00431-t001:** Primer designing for RxLR effectors with dEER motif.

Serial Number	Sequence ID	SequenceForward   Reverse	TM ^1^	AT ^2^
1	11472_g	 ATGAATAAGCGATATCTTTTGCCTTATAAACCAATCAATTAT 	4947	48
2	14151_g	ATGAAACAGATGATAAAAAGCTTAGCGTTTTGACTTTTTACC	4951	50
3	18087_g	ATGAACCCGAACATCATCTTCCTAGGGATTATTGACAAAGTA	5551	53
4	75485_g	ATGCGTCCTGTGTCCATCTTGTCACTTTGCAAAGCGTGAAAT	5953	56
5	7856_g	ATGAGACCGAACATTTTCTTCCTAGGGATTCTTGGCAAAGTA	5355	54
6	69274_g	ATGAACCCGAACATTGTATTCCTAGGGATTCTTTTTGGCAAA	5353	53
7	6877_g	ATGCGTCTCCGTCTGTTCCTTTACTCGCTCGCTTCACTGG	5858	58
8	10624_g	ATGCGCTATTCCATCCTTCTCTTAATGTTTTTTATACCAGTC	5747	52
9	60945_g	ATGCGGTCCGTCTCTATCCTCTTAGACTGGGTGGGCATTCTT	6157	59
10	8311_g	ATGCGGTCCGTCTCTATCCTTTAGACTGGGTGGGCATTCT	5856	57
11	35983_g	ATGATCTACGCCCCCTTAGTTTTTGAACGGGCAATGGTGTAG	5757	57
12	60741_g	ATGCGTTTCCATAGCCTGATTTTCCTCTGTAGCGGAAGCTCT	5559	57
13	38171_g	ATGGTCAACGCACTCTTGGTTTTAATTTACGCGACGTTTCTT	5751	54
14	46338_g	ATGATCAACAAACTCTTGGTTTTACGCGATTTTTCTTCTTTT	5149	50
15	74027_g	ATGATCCACAAACTCTTGGTTTTACGCGATTTTTCTTCTTTT	5349	51
16	10548_g	ATGAAGCTTTCTCTTCTCTTCTTACGGTGACATGAGCTTCCG	5359	56
17	1854_g	ATGAGAGCAACTTGTCTCCTATCACTGACGGTACCCGTTCTT	5559	57

^1^ TM: Melting Temperature, ^2^ AT: Annealing Temperature.

**Table 2 jof-09-00431-t002:** List of crinkler (CRN) proteins prognosticated by EffectorP tool.

Serial Number	Protein Number	Cytoplasmic Effector	Apoplastic Effector	Non-Effector	Prediction
1	681_g	Y (0.856)	-	-	Cytoplasmic effector
2	1492_g	-	-	Y (0.833)	Non-effector
3	1671_g	Y (0.823)	-	-	Cytoplasmic effector
4	3769_g	Y (0.785)	-	-	Cytoplasmic effector
5	5109_g	Y (0.799)	-	-	Cytoplasmic effector
6	5588_g	Y (0.89)	-	-	Cytoplasmic effector
7	8689_g	Y (0.856)	-	-	Cytoplasmic effector
8	8943_g	Y (0.903)	-	-	Cytoplasmic effector
9	10070_g	Y (0.89)	-	-	Cytoplasmic effector
10	10105_g	-	-	Y (0.735)	Non-effector
11	12349_g	Y (0.856)	-	-	Cytoplasmic effector
12	23122_g	Y (0.856)	-	-	Cytoplasmic effector
13	24510_g	Y (0.887)	-	-	Cytoplasmic effector
14	27640_g	Y (0.891)	-	-	Cytoplasmic effector
15	27641_g	Y (0.924)	-	-	Cytoplasmic effector
16	28426_g	Y (0.726)	-	-	Cytoplasmic effector
17	36964_g	Y (0.575)	-	-	Cytoplasmic effector
18	37717_g	Y (0.831)	-	-	Cytoplasmic effector
19	38162_g	Y (0.815)	-	-	Cytoplasmic effector
20	40025_g	Y (0.934)	-	-	Cytoplasmic effector
21	41242_g	Y (0.934)	-	-	Cytoplasmic effector
22	42778_g	-	-	Y (0.73)	Non-effector
23	47345_g	Y (0.523)	-	-	Cytoplasmic effector
24	49318_g	Y (0.785)	-	-	Cytoplasmic effector
25	50963_g	Y (0.729)	-	-	Cytoplasmic effector
26	53743_g	-	-	Y (0.768)	Non-effector
27	54388_g	Y (0.835)	-	-	Cytoplasmic effector
28	62219_g	Y (0.523)	-	-	Cytoplasmic effector
29	62440_g	Y (0.891)	-	-	Cytoplasmic effector
30	62442_g	Y (0.822)	-	-	Cytoplasmic effector
31	62857_g	Y (0.89)	-	-	Cytoplasmic effector
32	62858_g	Y (0.859)	-	-	Cytoplasmic effector
33	68815_g	Y (0.813)	-	-	Cytoplasmic effector
34	70580_g	Y (0.582)	-	-	Cytoplasmic effector
35	72237_g	Y (0.824)	-	-	Cytoplasmic effector

**Table 3 jof-09-00431-t003:** List of RxLR proteins predicted by Effector P tool.

Serial Number	Protein Number	Cytoplasmic Effector	Apoplastic Effector	Non-Effector	Prediction
1	21379_g	Y (0.959)	-	-	Cytoplasmic effector
2	3464_g	Y (0.809)	-	-	Cytoplasmic effector
3	47557_g	Y (0.905)	-	-	Cytoplasmic effector
4	12504_g	-	-	Y (0.755)	Non-effector
5	55207_g	Y (0.791)	-	-	Cytoplasmic effector
6	25335_g	Y (0.843)	-	-	Cytoplasmic effector
7	11898_g	-	-	Y (0.81)	Non-effector
8	35685_g	Y (0.903)	-	-	Cytoplasmic effector
9	59897_g	-	-	Y (0.8)	Non-effector
10	63043_g	-	-	Y (0.573)	Non-effector
11	9023_g	Y (0.84)	-	-	Cytoplasmic effector
12	18507_g	Y (0.736)	-	-	Cytoplasmic effector
13	65651_g	Y (0.685)	-	-	Cytoplasmic effector
14	10686_g	Y (0.906)	-	-	Cytoplasmic effector
15	45770_g	Y (0.868)	-	-	Cytoplasmic effector
16	26637_g	Y (0.876)	-	-	Cytoplasmic effector
17	46575_g	Y (0.761)	-	-	Cytoplasmic effector
18	1289_g	Y (0.905)	-	-	Cytoplasmic effector
19	53288_g	Y (0.955)	-	-	Cytoplasmic effector
20	10613_g	Y (0.832)	-	-	Cytoplasmic effector
21	20048_g	Y (0.723)	-	-	Cytoplasmic effector
22	23346_g	Y (0.832)	-	-	Cytoplasmic effector
23	35739_g	Y (0.806)	-	-	Cytoplasmic effector
24	44650_g	Y (0.827)	-	-	Cytoplasmic effector
25	60111_g	Y (0.903)	-	-	Cytoplasmic effector
26	70669_g	Y (0.86)	-	-	Cytoplasmic effector
27	7606_g	Y (0.827)	-	-	Cytoplasmic effector
28	9639_g	Y (0.832)	-	-	Cytoplasmic effector
29	22687_g	Y (0.883)	-	-	Cytoplasmic effector
30	24069_g	Y (0.921)	-	-	Cytoplasmic effector
31	29669_g	Y (0.883)	-	-	Cytoplasmic effector
32	35014_g	Y (0.964)	-	-	Cytoplasmic effector
33	39202_g	Y (0.87)	-	-	Cytoplasmic effector
34	62695_g	Y (0.964)	-	-	Cytoplasmic effector
35	60583_g	Y (0.569)	-	-	Cytoplasmic effector
36	68703_g	Y (0.571)	-	-	Cytoplasmic effector
37	71584_g	Y (0.96)	-	-	Cytoplasmic effector
38	13581_g	Y (0.808)	-	-	Cytoplasmic effector
39	34223_g	Y (0.894)	-	-	Cytoplasmic effector
40	37513_g	Y (0.759)	-	-	Cytoplasmic effector
41	77025_g	Y (0.731)	-	-	Cytoplasmic effector
42	29984_g	Y (0.746)	-	-	Cytoplasmic effector
43	42066_g	Y (0.872)	-	-	Cytoplasmic effector
44	42069_g	Y (0.945)	-	-	Cytoplasmic effector
45	32234_g	Y (0.915)	-	-	Cytoplasmic effector
46	28623_g	Y (0.927)	-	-	Cytoplasmic effector
47	32891_g	Y (0.845)	-	-	Cytoplasmic effector
48	1182_g	Y (0.88)	-	-	Cytoplasmic effector
49	17918_g	Y (0.803)	-	-	Cytoplasmic effector
50	19540_g	Y (0.64)	-	-	Cytoplasmic effector
51	64364_g	-	-	Y (0.781)	Non-effector
52	75770_g	Y (0.796)	-	-	Cytoplasmic effector

**Table 4 jof-09-00431-t004:** List of RxLR-dEER proteins prognosticated by Effector P tool.

Serial Number	Sequence ID	Cytoplasmic Effector	Apoplastic Effector	Non-Effector	Prediction
1	10548_g	Y (0.7)	-	-	Cytoplasmic effector
2	14151_g	Y (0.927)	-	-	Cytoplasmic effector
3	60741_g	Y (0.908)	-	-	Cytoplasmic effector
4	75485_g	Y (0.946)	-	-	Cytoplasmic effector
5	11472_g	Y (0.893)	-	-	Cytoplasmic effector
6	6877_g	Y (0.898)	-	-	Cytoplasmic effector
7	1854_g	-	-	Y (0.619)	Non-effector
8	35983_g	Y (0.8)	-	-	Cytoplasmic effector
9	38171_g	Y (0.88)	-	-	Cytoplasmic effector
10	46338_g	Y (0.908)	-	-	Cytoplasmic effector
11	74027_g	Y (0.908)	-	-	Cytoplasmic effector
12	7856_g	Y (0.585)	Y (0.501)	-	Cytoplasmic/apoplastic effector
13	18087_g	Y (0.816)	-	-	Cytoplasmic effector
14	69274_g	Y (0.799)	-	-	Cytoplasmic effector
15	60945_g	Y (0.921)	-	-	Cytoplasmic effector
16	8311_g	Y (0.92)	-	-	Cytoplasmic effector
17	10624_g	Y (0.938)	-	-	Cytoplasmic effector

**Table 5 jof-09-00431-t005:** List of novel RxLR-dEER gene sequences.

Serial Number	Sequence ID	Gene Bank Accession Number	Size in Base Pairs
1	35983_g	OM365913	1410
2	6877_g	OM135515	885
3	8311_g	OM365912	1254
4	60945_g	OM365911	1248
5	60741_g	OM365914	1533

**Table 6 jof-09-00431-t006:** Percentage of overall intrinsic disorder of the amino acid sequences.

Band Number	Sequence ID	Gene Bank Accession Number	Overall Disorder (%)
7	6877_g	OM135515	43.88
9	60945_g	OM365911	26.75
10	8311_g	OM365912	28.06
11	35983_g	OM365913	46.17
12	60741_g	OM365914	25.05

**Table 7 jof-09-00431-t007:** Sequence features and predicted domains of the effector proteins.

Serial Number	Protein Name	Signal Peptide Likelihood	Cleavage Site between Position	RxLR Motif Position	DEER Motif Position
1	6877_g	0.9996	16 and 17	40 to 43	53 to 56
2	60945_g	0.9997	21 and 22	48 to 51	58 to 61
3	8311_g	0.9997	21 and 22	48 to 51	58 to 61
4	35983_g	0.9998	19 and 20	43 to 46	55 to 58
5	60741_g	0.9998	19 and 20	43 to 46	56 to 59

## Data Availability

The data used to support the results of this paper is available at the NCBI repository with accession numbers OM135515, OM365911, OM365912, OM365913, and OM365914 is accessible through the following link, https://www.ncbi.nlm.nih.gov/nucleotide/ (accessed on 27 August 2022). *Sclerosporagraminicola* genome available in NCBI GenBank used for transcriptome mapping: https://www.ncbi.nlm.nih.gov/bioproject/PRJNA325098/ (accessed on 27 August 2022).

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
