# Peer review of "Genome-Wide Characterization of Effector Protein-Encoding Genes in Sclerospora graminicola and Its Validation in Response to Pearl Millet Downy Mildew Disease Stress"

_jof, 2023, doi:10.3390/jof9040431_

Round 1
Reviewer 1 Report
In the manuscript (ms) entitled “Genome wide characterization of effector protein-encoding genes in Sclerospora graminicola and its validation in response to pearl millet downy mildew disease stress”. Authors received fascinating results on the integration of effectors in oomycetes pathogens and particularly their involvement in the host plant, and were able to logically interpret their data. This Reviewer has some comments that should be addressed to improve the presentation and readability of the ms:
1. The abstract reflects what the authors found in this study. Author may include a sentence at the end of the summary (conclusion) whether this study would provide any useful advice to the readers?
2. Abstract: I suggest to elaborate the crinkler and RXLR groups.
3. Please delete, (PCR, gel-electrophoresis) in the abstract
4. Clarify how the gene expression is done. To me it looks gene validation? confirm.
5. The word “undoubtedly’ is vague to use in research.
6. In the introduction, what is effectome?
7. The first usage should be expanded and followed by abbreviations.
8. Were the softwares used for analysis are source of your institute?
9. Remove the hyperlink of the softwares in the whole ms.
10. Explain the basis of primer synthesis.
11. What is CRN motif. Is it have a role in the effector?
12. Many places words are merged. Please re-check.
13. Discussion: The paper will be more attractive, if the authors discuss the role of effectors in other cereal crops pathogen interaction.
Author Response
Responses to the Reviewers’ Comments
The authors would like to thank the Editor and Reviewer’s for his/her constructive comments and suggestions that have helped us improve our manuscript. The detailed response to each comment is listed below by point to point. In addition, an extensive revision has been undertaken and incorporated all the corrections and suggestions raised by the Reviewer’s in the revised manuscript.
Reviewer# 1
Comment 1: The abstract reflects what the authors found in this study. Author may include a sentence at the end of the summary (conclusion) whether this study would provide any useful advice to the readers?
Response: We highly appreciate this reviewer for raising this query. We have now incorporated briefly the overall outcome/ conclusion of the study which we believe will reach wider audience.
“These results will assist in identifying functional effector proteins involving omic approach using newer bioinformatic tools to protect pearl millet plants against downy mildew stress. Taken together, the identified effector protein-encoding functional genes can be utilized in screening oomycetes downy mildew diseases in other crops across the globe.” (Page 1).
Comment 2: Abstract: I suggest to elaborate the crinkler and RXLR groups.
Response: Thank you so much for this comment. Accordingly, we have elaborated the abbreviation in its first usage in each section of the revised manuscript (Page 1).
Comment 3: Please delete, (PCR, gel-electrophoresis) in the abstract
Response: Thank you, the suggested word/s are deleted from the revised manuscript.
Comment 4: Clarify how the gene expression is done. To me it looks gene validation? confirm.
Response: Thank you so much for this critical observation of our manuscript. We are extremely sorry for this mistake. The present work emphasizes on gene validation only, hence appropriate correction is made in the revised manuscript (Page 1).
Comment 5: The word “undoubtedly’ is vague to use in research.
Response: We totally agree with your suggestion. The word “undoubtedly’ is now deleted and the entire sentence is now revised as follows:
“These results will assist in identifying functional effector proteins involving omic approach using newer bioinformatic tools to protect pearl millet plants against downy mildew stress. Taken together, the identified effector protein-encoding functional genes can be utilized in screening oomycetes downy mildew diseases in other crops across the globe.” (Page 1).
Comment 6: In the introduction, what is effectome?
Response: The word effectome is synonym of effectorome as describe by Arroyo-Velez et al. (2020), which is the repertoire of all its effector proteins in a single pathogen (Page 2).
“Arroyo-Velez, N.; González-Fuente, M.; Peeters, N.; Lauber, E.; Noël, L. D. From Effectors to Effectomes: Are Functional Studies of Individual Effectors Enough to Decipher Plant Pathogen Infectious Strategies? PLOS Pathog. 2020, 16 (12), e1009059. https://doi.org/10.1371/journal.ppat.1009059”.
Comment 7: The first usage should be expanded and followed by abbreviations.
Response: Yes, the necessary correction/s has been done throughout the revised manuscript.
Comment 8: Were the softwares used for analysis are source of your institute?
Response: The softwares used in this study are not owned by our University.
Comment 9: Remove the hyperlink of the softwares in the whole ms.
Response: Thank you so much, as per your recommendation all the hyperlinks of the softwares were removed in the revised manuscript.
Comment 10: Explain the basis of primer synthesis.
Response: We highly appreciate the Reviewer for giving this important idea. To achieve the full length of effector protein-encoding nucleotide sequence from the genome, we designed the primers manually by selecting few nucleotides from the site of initiation and from the site of termination, a few nucleotides were selected based on reverse complement tool (https://www.bioinformatics.org/sms/rev_comp.html). To calculate melting temperature of the primer we have used percent GC Oligocalc tool (https://justbio.com/oligocalc/).
Comment 11: What is CRN motif. Is it have a role in the effector?
Response: Thank you so much again for this comment. We have mistakenly written as LxLFLAK motif as CRN motif. It is been reported that these motifs are involved in translocation of effector proteins from pathogen into the host (Schornack et al., 2010) and in the present work effectors are categorized based on the motif’s present at the N terminal of the amino acid sequence.
Schornack, S.; van Damme, M.; Bozkurt, T. O.; Cano, L. M.; Smoker, M.; Thines, M.; Gaulin, E.; Kamoun, S.; Huitema, E. Ancient Class of Translocated Oomycete Effectors Targets the Host Nucleus. Proc. Natl. Acad. Sci. 2010, 107 (40), 17421–17426. https://doi.org/10.1073/pnas.1008491107.
Comment 12: Many places words are merged. Please re-check.
Response: Sorry for this error/s. All such errors have been fixed properly in the revised manuscript.
Comment 13: Discussion: The paper will be more attractive, if the authors discuss the role of effectors in other cereal crops pathogen interaction.
Response: We would like to express our special thanks to the Reviewer for his / her comment. To meet the reviewer’s comment, the role of effectors in other cereal crops is incorporated in the revised manuscript. For instance, Hacquard et al. (2013) discovered a multistep mode of action of mildew candidate secretory effector proteins (CSEPs) in powdery mildew pathogenesis on barley and immunocompromised Arabidopsis, with a first wave of CSEP transcripts accumulating during host cell entry (12 hour) and a second wave of transcripts accumulating at the stage of haustorium formation (24 hour). In wheat powdery mildew, a comparable high induction of CSEPs was seen during the haustorial stage (Praz et al., 2018) (Page 14).
Hacquard S, Kracher B, Maekawa T, Vernaldi S, Schulze-Lefert P, Ver Loren van Themaat E: Mosaic genome structure of the barley powdery mildew pathogen and conservation of transcriptional programs in divergent hosts. Proc Natl Acad Sci 2013, 110:E2219-E2228.
Praz CR, Menardo F, Robinson MD, Muller MC, Wicker T, Bourras S, Keller B: Non-parent of origin expression of numerous effector genes indicates a role of gene regulation in host adaption of the hybrid triticale powdery mildew pathogen. Front Plant Sci 2018, 9:49.

Reviewer 2 Report
This study has established a pipeline for identifying novel effector candidates from an obligate biotrophic fungal pathogen, Sclerospora graminicola, and presents important insights on finding novel effectors when infecting pearl millet. However, as a high-impact research journal, I expect to see more informative results from the effectors that the authors have identified.
The results lack in-depth in-silico or genomics analysis of the effectors in terms of their sequence features, predicted domains and functions, phylogenetic relationships to other RxLR or CRN effectors that have been identified from other pathogens. Such comparison or analysis would provide a more comprehensive understanding of the uniqueness of the effectors from S. graminicola.
Besides, the study lacks functional validation results (in vivo or in vitro) of the identified effectors. Experiments could include gene expression profiling during disease development, plant immune responses to effectors when transiently expressed, or how these effectors cause disease or are recognized by plants. Such validation results would provide more information on the virulent roles of the identified effectors.
I acknowledge the authors’ contribution to mining unique effectors from an obligate fungal pathogen, which have undoubtedly faced some technical difficulties. However, given the high standards and expectations of our journal, I suggest that the authors could conduct additional experiments to address the niche of the identified effectors regarding to their evolutionary or virulent roles for S. graminicola. This information will strengthen this study and provide more insighst on S. graminicola pathogenicity.
Author Response
Responses to the Reviewers’ Comments
The authors would like to thank the Editor and Reviewer’s for his/her constructive comments and suggestions that have helped us improve our manuscript. The detailed response to each comment is listed below by point to point. In addition, an extensive revision has been undertaken and incorporated all the corrections and suggestions raised by the Reviewer’s in the revised manuscript.
Reviewer #2
Comment 1: This study has established a pipeline for identifying novel effector candidates from an obligate biotrophic fungal pathogen, Sclerospora graminicola, and presents important insights on finding novel effectors when infecting pearl millet. However, as a high-impact research journal, I expect to see more informative results from the effectors that the authors have identified.
Response: We would like to express our special thanks to this Reviewer for positively evaluating our manuscript and also providing constructive comments which will help us to improve the quality of our manuscript. We have addressed each of the comment very objectively and the responses for the individual comments are as below:
Comment 2: The results lack in-depth in-silico or genomics analysis of the effectors in terms of their sequence features, predicted domains and functions, phylogenetic relationships to other RxLR or CRN effectors that have been identified from other pathogens. Such comparison or analysis would provide a more comprehensive understanding of the uniqueness of the effectors from S. graminicola.
Response: Thank you very much for this critical observation of our manuscript. Following your recommendation, we have characterized the effector protein using signalP 6.0 tool for better understanding the involvement of sequence features and predicted domains of the five novel effector protiens in S. graminicola and the data are presented in the figure 3.1 and table 7 (Page 12, Please see below). In addition, we have carried out the comparison studies using amino acid sequence and generated the phylogenetic tree showing relationships to other proteins of doany mildew causing pathogens (Supplementary Table 1 and Figures S1-5).
Comment 3: Besides, the study lacks functional validation results (in vivo or in vitro) of the identified effectors. Experiments could include gene expression profiling during disease development, plant immune responses to effectors when transiently expressed, or how these effectors cause disease or are recognized by plants. Such validation results would provide more information on the virulent roles of the identified effectors.
I acknowledge the authors’ contribution to mining unique effectors from an obligate fungal pathogen, which have undoubtedly faced some technical difficulties. However, given the high standards and expectations of our journal, I suggest that the authors could conduct additional experiments to address the niche of the identified effectors regarding to their evolutionary or virulent roles for S. graminicola. This information will strengthen this study and provide more insighst on S. graminicola pathogenicity.
Response: We highly appreciate this Reviewer for his critical evaluation of our manuscript. According to the Reviewer suggestion, the sequence features and predicted domains are included in the revised manuscript. In this study, all the five RxLR have both the motifs (RxLR and DEER) after the cleavage site at N-terminals of the amino acid sequences and C-terminal regions are highly conserved helical structures, it is been reported that these motifs are involved in translocation of effectors into the host.
The sequence features and predicted domains of the five novel effector protiens are presented in the figure 3.1 and table 7. In addition, all the five RxLR effector protein encoding genes doesn’t have much similarities in the existing database of NCBI, hence constructing a phylogenetic is not possible. However, the amino acid sequence showed similarity and the phylogeny tree was drawn and incorpaorted as supplementary figure (Supplementary table 1 and Figures S5) .
Figure 3.1. Prediction of signal peptide and likelihood by SignalP-6.0, where N-terminal n-region, the hydrophobic h-region and the C-terminal c-region. (A). Protein 6877- One Secretory signal peptide and cleavage site at 16th position with likelihood of 0.9996 and probability of 0.536864. (B). Protein 60945- One signal peptide and cleavage site at 21st position with likelihood of 0.9997 and probability of 0.941781. (C). Protein 8311- One signal peptide and cleavage site at 21st position with likelihood of 0.9997 and probability of 0.943846. (D). Protein 35983- One signal peptide and cleavage site at 19th position with likelihood of 0.9998 and probability of 0.974416. (E). One signal peptide and cleavage site at 19th position with likelihood of 0.9998 and probability of 0.966619.
Furthermore, in the present work 35983_g gene was found to have highest number of disordered residues, hence it was selected for gene expression studies. This amplified gene was cloned using the plasmid vector of bacterial expression system. The protein from was extracted from the bacteria and was spry inoculated on the 16 days old callus of pearl millet to check the histochemical changes when compared to untreated control. We found that the effector protein spray inoculated callus had developed hypersensitivity reaction (HR) in the form of dark spots which were visible at 2 hours post inoculation when compared to control (data not shown in manuscript).
Figure S1. Observation of histochemical changes on callus under 40X magnification of bright field microscope (A). No HR was observed on untreated callus. (B). Appearance of necrotic spots and streaks after 2 hours of spry inoculation of effector protein on the callus.

Reviewer 3 Report
The manuscript entitled "Genome wide characterization of effector protein-encoding 2 genes in Sclerospora graminicola and its validation in response 3 to pearl millet downy mildew disease stress" is an interesting topic of research. The work is well organized and the conclusions are supported by the results. The manuscript presents important tools for designing new strategies for protecting plants against downy mildew stress and in this way, can be from the interest of the JoF readers. I have only a minor point regarding the Figure 3. Authors can prepare a figure with only one caption, this mean, Figure 3: A)... B).. C)... Also, maybe the data presented in the figure could be presented in a supplementary material. This suggestion can be done during the proof step.
So, based on my comments, I recommend the acceptance of the manuscript.
Author Response
Responses to the Reviewers’ Comments
The authors would like to thank the Editor and Reviewer’s for his/her constructive comments and suggestions that have helped us improve our manuscript. The detailed response to each comment is listed below by point to point. In addition, an extensive revision has been undertaken and incorporated all the corrections and suggestions raised by the Reviewer’s in the revised manuscript.
Reviewer #3
Comment 1: The manuscript entitled "Genome wide characterization of effector protein-encoding 2 genes in Sclerospora graminicola and its validation in response to pearl millet downy mildew disease stress" is an interesting topic of research. The work is well organized and the conclusions are supported by the results. The manuscript presents important tools for designing new strategies for protecting plants against downy mildew stress and in this way, can be from the interest of the JoF readers. I have only a minor point regarding the Figure 3. Authors can prepare a figure with only one caption, this mean, Figure 3: A)... B).. C)... Also, maybe the data presented in the figure could be presented in a supplementary material. This suggestion can be done during the proof step.
Response: We would like to thank this reviewer for his valuable comments which we believe will improve the quality of our manuscript. We have taken care of all such error/s. In addition, the suggested changes in legends of Fig. 3 have been made in the revised manuscript
“Figure 3. (A). 35983_g with overall 46.17% disordered amino acid residues. (B). 6877_g with overall 43.88% disordered amino acid residues. (C). 8311_g with overall 28.06% disordered amino acid residues. (D). 60945_g with overall 26.75% disordered amino acid residues. (E). 60741_g with overall 25.05% disordered amino acid residues” (Page 11).

Round 2
Reviewer 2 Report
Comments to the authors:
I appreciate the revision from the authors, and I acknowledge the authors’ efforts to response to my comments timely and properly. I think this revised version of manuscript is more informative and makes the story complete.
I have only some minor suggestions:
1. Please rephrase the sentence in line 241-243. It’s not clear about what you are trying to say here.
2. Change the title for section 3.5. “Gene expression analysis”. It’s not really a gene expression analysis through qPCR or relative RT-PCR. It’s a confirmation of the presence of the genes.
3. Line 279, change Figure 3.1 to Figure 4 and rephrase the figure caption.
4. Figure S1 to S5: use high quality figures. It’s not clear neither on screen or on paper.
5. For the HR results, I suggest just described this results in your discussion but not put it in the supplementary data. It will require details on how to express and purify the effector from bacteria (E. coli?) and need western blot results to see the presence of the effector. It will be a strong result if providing with proper controls but it might take couples of months to show the complete results. I will suggest just simply describing this finding in your discussion that this effector can potentially cause cell death.
6. Please be aware of some language issues like tense or singular and plural.
Author Response
I appreciate the revision from the authors, and I acknowledge the authors’ efforts to response to my comments timely and properly. I think this revised version of manuscript is more informative and makes the story complete.
Response: We would like to thank this reviewer for his positive evaluation of our manuscript.
I have only some minor suggestions:
Comment 1: Please rephrase the sentence in line 241-243. It’s not clear about what you are trying to say here.
Response: Thank you so much for your critical observation of our manuscript. To avoid confusion, the sentence is now revised as follows:
“However the translated RxLR protein sequence showed similarity with other proteins found in the NCBI database (Supplementary table 1; Supplementary Figure S1-S5)”.
Comment 2: Change the title for section 3.5. “Gene expression analysis”. It’s not really a gene expression analysis through qPCR or relative RT-PCR. It’s a confirmation of the presence of the genes.
Response: As per your recommendation the sub-tile is revised as below:
“3.5. Confirmation of the presence of RxLR-dEER effectors genes”
Comment 3: Line 279, change Figure 3.1 to Figure 4 and rephrase the figure caption.
Response: We have replaced Figure 3.1 to Figure 4 and the legend is revised as follows:
“Figure 4. Effector proteins representing the secretary signal peptide and its cleavage site generated using SignalP 6.0”
Comment 4: Figure S1 to S5: use high quality figures. It’s not clear neither on screen or on paper.
Response: Thank you so much for this comment. We have supplied a improved version of supplementary figure S1 to S5 with high resolution.
Comment 5: For the HR results, I suggest just described this results in your discussion but not put it in the supplementary data. It will require details on how to express and purify the effector from bacteria (E. coli?) and need western blot results to see the presence of the effector. It will be a strong result if providing with proper controls but it might take couples of months to show the complete results. I will suggest just simply describing this finding in your discussion that this effector can potentially cause cell death.
Response: We would like to express our special thanks to this reviewer for raising this important comment and suggestion. To meet the Reviewer suggestion, we have now incorporated the below sentence in discussion section of the revised manuscript.
“In this study, the hypersensitive reaction was indicated by the development of necrotic areas on the resistant pearl millet callus within 2 hour post inoculation, thus triggering defense signaling responses in the neighbouring cells”.
Comment 6: Please be aware of some language issues like tense or singular and plural.
Response: To meet the Journal quality, we have taken care all our efforts to improve the english and scientific content of the manuscript.
